# One-Layer vs. Two-Layer SOM in the Context of Outlier Identification: A Simulation Study

**Gabriel Antonio Valverde Castilla** [1,*] **, José Manuel Mira McWilliams** [2] **and Beatriz González-Pérez** [1,3]

1 Department of Statistics and Operations Research, Faculty of Mathematics, Complutense University of Madrid, 28040 Madrid, Spain; beatrizg@ucm.es
2 Organization Engineering, Business, Administration and Statistics, Technical School of Industrial Engineering, Universidad Politécnica de Madrid, 28006 Madrid, Spain; josemanuel.mira@upm.es
3 Interdisciplinary Mathematics Institute (IMI), Complutense University of Madrid, 28040 Madrid, Spain
* Correspondence: gvalverd@ucm.es; Tel.: +34-600-442-627

**Abstract:** In this work, we applied a stochastic simulation methodology to quantify the power of the detection of outlying mixture components of a stochastic model, when applying a reduced-dimension clustering technique such as Self-Organizing Maps (SOMs). The essential feature of SOMs, besides dimensional reduction into a discrete map, is the conservation of topology. In SOMs, two forms of learning are applied: competitive, by sequential allocation of sample observations to a winning node in the map, and cooperative, by the update of the weights of the winning node and its neighbors. By means of cooperative learning, the conservation of topology from the original data space to the reduced (typically 2D) map is achieved. Here, we compared the performance of one- and two-layer SOMs in the outlier representation task. The same stratified sampling was applied for both the one-layer and two-layer SOMs; although, stratification would only be relevant for the two-layer setting—to estimate the outlying mixture component detection power. Two distance measures between points in the map were defined to quantify the conservation of topology. The results of the experiment showed that the two-layer setting was more efficient in outlier detection while maintaining the basic properties of the SOM, which included adequately representing distances from the outlier component to the remaining ones.

**Keywords:** self-organizing map; neural networks; robustness; nonlinear projections; dimensionality reduction; deep semisupervised learning

## 1. Introduction

The purpose of this paper was to apply stochastic simulation for a better understanding of the possibilities of outlier component detection in a Gaussian mixture using one- and two-layer Self Organizing Maps (SOMs). SOMs were developed by Kohonen [1] as a tool to represent structures such as cortical layers in the brain as two- or three-dimensional maps. In more abstract terms, a SOM is a clustering technique that implies a reduction of the dimensionality to two or three dimensions, thus providing a visual description of the clustering. A priori, SOMs need not be related to a stochastic reality: it can be used within a deterministic framework. The first reference to SOMs was the seminal paper by Kohonen [1], based on the work by von der Malsburg [2] and Willsh and von Der Malsburg [3] on competitive learning. The theoretical background was in the monograph by Kohonen [4]. Two interesting and thorough reviews of SOMs can be found in Yin [5] and Van Hulle [6]. The stochastic approaches started with Lutrell [7] and Yin and Alison [8]; a more recent reference was Guo et al. [9]. Deep SOMs were approached from a practical of view by Liu et al. [10]. The relationship between SOMs and the EM algorithm was explored by Yuille [11], Durbin et al. [12], and Utsugi, A. [13]. Mixtures within the Bayesian framework were dealt with by Lau and Green, P. [14], Wang and Dunson, D. [15], and Ormoneit and Tresp, Dahl [16].

Here, we formulated a stochastic data generating process by means of a mixture of Gaussian distributions, where one of the mixture components is a low-weight outlier. Given that in many real problems, it is interesting to keep a representation of the outlier in the map while respecting the essence of the standard SOM, we studied how the SOM is able to do so in some situations as simulated in our experiment and checked if our sense that two-layer SOMs would do better in such outlier representations and detection than one-layer ones, as well as an adequate "between outlier component and remaining component distance" representation, was correct. This would be achieved without penalizing an adequate representation of the nonoutlier components.

To this end, we generated a stratified sample, in such way that we mixed in one stratum the outlying component and the one closest to it.

Then, for the one-layer SOM, where the stratification is irrelevant, the outlying component would be diluted in the node corresponding to the stratum it shares with the neighboring (closest) component (due to the difference in frequencies, the statistic is more biased towards the closest component), or the nodes that could be seen as representative of the different components appear abruptly disconnected from each other, so distances are less well preserved. Due to the low frequency of the outlying component and its strong deviation with respect to the remaining components, there will be very few nodes associated with this component, and they will be updated very few times since there are few sample points belonging to that component. Its neighbors, on the contrary, will be updated much more often because they are far apart in the map and also because they will the winning nodes for other (no outlier) sample points.

In the two-layer SOM, in the first intermediate layer, we built a map for each stratum and then used the nodes of these first-layer SOMs as the data for the second (and last layer) map. Then, since the stratum including the outlier will have several nodes associated with it in its first layer map, one of these nodes may adequately represent the outlying component. If so, it may receive a node of its own in the second (final) map. This way, it would have a higher chance of being represented than in a single-layer SOM. Thus, the two-layer model made the difference when we work with low-frequency components of very extreme values, because it provided more representative nodes in the intermediate layers.

The SOM is a tradeoff between efficient clustering, i.e., achieving the least possible intracluster (node) variance, at the cost of the largest possible between-cluster variability, and at the same time the conservation of topology. In this work, we added to these two optimality criteria the additional one of the isolation/detection of the outlier component, and the degree of this isolation/detection could be measured by the mean distance from the outlier component sample elements' representations in the map (the weight of their winning node) to the center of the original component. One can thus define an optimal tradeoff among these three criteria. Each stratification will perform better or worse for each of the criteria, and on whichever tradeoff between these three criteria is defined. The two-layer should not sacrifice the two other criteria in favor of outlier isolation; it should also perform well as far as the two former (criteria) are concerned.

An important question here is how one measures the conservation of topology, i.e., which is the SOM distance between two points $x_i$ and $x_j$ in the original space. Note that when comparing two SOMs in the preservation of topology, the distance in the original space is the same for both, so one just has to compare the distances in the SOM. Should it just depend on the integer value map coordinates or also on the corresponding winning node weights (centroids)? The approach of this paper was to take both into account when defining a SOM distance. This issue is treated in detail in Section 4.

Additionally, the stratification chosen should not be the result of using too much prior information on the position of the outlier.

*Contributions*

The main contributions that we describe in this work are: 1. The two-layer structure for more efficient detection of outlier clusters. 2. The two between-maps distance measures: one image-based and the second graph-based with the propose of quantifying how the original topology is preserved. 3. A simulation example to illustrate how the two-layer strategy is more efficient. This required a stratification of the simulation sample. 4. Some real-world examples where this outlier detection is useful and relevant.

The contents of the paper are structured as follows: Section 2 reviews the SOM sequential algorithm. Section 3 describes the simulation experiment, specifying the stochastic model, sampling strategy, and one- and two-layer SOM details. Section 4 is devoted to the conservation of topology. The results and discussion are shown in Sections 5 and 6.

## 2. The SOM Algorithm

We now present a short review of the original SOM algorithm.

The SOM is a neural network that allows us to project a high-dimensional vector space onto a low-dimensional topology (typically two) integrated by a set of different nodes or neurons displayed as a grid. This nonlinear projection results in a "feature map" of the high-dimensional space, which may be used to detect and analyze characteristics that allow for the identification of patterns or groups. SOMs have been applied to a number of fields, e.g., document retrieval, financial data analysis, forensic analyses, and engineering applications. Machine learning includes two categories, supervised and unsupervised learning. In the former, there is a division of the variables between inputs and outputs, and the purpose of the analysis is to estimate the input–output relationship and make predictions. In the latter, all variables are on equal footing, and one searches for a better understanding of their multivariate structure, possibly also involving dimensional reduction. With (unsupervised) SOMs, we are capable of identifying features and structures in high-dimensional data. SOMs show a self-organizing behavior with the capacity to detect hidden characteristics within nonlabeled groups when enough map nodes (also called neurons) are used. It also has the perspective of dimensional reduction that seeks to optimize topological conservation [1,17,18], that is that the relative locations of the points in the original space are reflected in some way on the map. One of those ways is distance preservation. Thus, it can be used to identify similar objects once the map has been trained.

As mentioned above, the SOM produces a nonlinear mapping of the high-dimensional space onto a reduced dimension one. Several versions of the original algorithm [1,7] have been proposed. Suppose $x(t)$ denotes the input vectors and the weight vector of the $k$-th neuron (node) ($k = 1...K$). The $K$ neurons are arranged in a bidimensional network, which induces a set of corresponding weight vectors. A discrete time (training epoch) index ($t = 1...$) is introduced so that $x(t)$ is presented to the network at time $t$, and $w_k(t)$, which represents the state of the net at that moment, are updated. Before training the map, the weight vectors should be initialized, which is generally performed by taking K vectors, typically by random sampling within the initial data.

Once the map has been trained, any new observation will be assigned to the map node with the lowest euclidean distance between the weight and observation (obviously, in the original space).

*2.1. Sequential (Original) SOM*

In the original, so-called sequential SOM, observations are introduced one at a time, as opposed to the batch SOM.

For the training phase, a metric is required for the distance between vectors in the input space, and the euclidean distance is typically applied when the variables are continuous and on equal scale [1]. For each input vector introduced, the closest (euclidean distance) weight vector is obtained. The so-called winning vector is denoted by the $c$ subindex. Finally, all weight vectors are updated according to Equation (1).

$$\omega_k(t+1) = \omega_k(t) + \alpha(t)h_{ck}(t)|x(t) - \omega_k(t)| \tag{1}$$

The weight updating process is mainly driven by the learning rate, which should decrease monotonically along time. A standard option is an exponential decrease:

$$\alpha(t) = \alpha_1 e^{-\frac{t}{\alpha_j}}, \tag{2}$$

where $\alpha_1$ and $\alpha_j$ are constants. The neighborhood function $h_{ck}(t)$ provides the conservation of topology through weight updating, usually called the neighborhood Equation (3) function. It thus defines the region—around the winning node—affected by the updating process. It controls the extent to which an input vector adjusts the weight of the neuron $k$ with respect to the winner $c$ by means of the corresponding (in the map) intra-net distance between the two vectors. Moreover, the $x(t) - \omega_k(t)$ term produces a stronger update—within nodes equally distant in the 2D map from the winning one—for that whose weight in the previous iteration is more distant from the new observation. Several functions have been proposed in this direction, and here, we applied a Gaussian neighborhood. This function is symmetric about the winner. Furthermore, it monotonically decreases with time due to $\sigma(t)$ and with distance to the winner.

$$h_{ck}(t) = exp(-|\omega_c - x_k|^2/\sigma^2(t)) \tag{3}$$

$$\sigma(t) = \sigma_1 e^{-\frac{t}{\sigma_j}} \tag{4}$$

---

**Algorithm 1** SOM algorithm.

---

SOM 1-layer

**Require:** $K \geq 0 \vee S \subset X$ sample $\vee$ epochs number
**Ensure:** $\omega_k \, \forall K$
  Initialize $\omega_k \, \forall k \in (1...K)$ randomly
  **for** all epoch **do**
    {random loop over all input vector $x \in X$}
    **for** each x **do**
      {update weight vector $\omega_k$ according to eq. 1}
      Note: Including winner and neighborhood function
    **end for**
    Note: Not update $\alpha$ and $\sigma$ because are function of the epoch
  **end for**

---

### 2.2. Two-Layer SOM

The SOM algorithm developed belongs to or is included within the unsupervised algorithms. Its objective is to project an original space into a two-dimensional space. This projection is not chaotic, but is governed by a main objective, to maintain the topology, to maintain the positional structure (except for rotations) of the points in the original space on the map [1,17,18]. This can be landed on keeping distances from the original space on the map.

However, there are a significant number of studies that have sought different strategies given the perspective that the model could be seen as too biased by the distribution and structure of the data [19,20].

Our proposal was to apply an assembled SOM, linking a single SOM, different maps generated for random or non-random strata of the data. This allowed us to provide a parallelizable solution while optimizing the process of the detection and representation

of low-frequency areas that we sought to be overrepresented in any of the blocks. We call them two-layer SOM. Algorithms 1 and 2.

The process therefore consisted of partitioning the initial dataset into n strata, applying the original SOM algorithm, the one-layer SOM, on each of them, and this was the parallelizable part. We generated a new dataset with all the weights generated in these maps and represented them in a new two-dimensional map.

---

**Algorithm 2** Two-layer SOM algorithm

---

**Require:** $m, n, p \in \mathbb{N}\; K \geq 0 \vee S \subset X$ sample $\vee$ epochs number
**Ensure:** $\omega_k^i\; \forall K$
   Split $S = S_1 \cup ... \cup S_p$
   **for** $S_i\; i \in (1, ..., p)$ **do**
      Initialize $\omega_k\; \forall k \in (1...K)$ randomly
      **for** all epoch **do**
         {random loop over all input vector $x \in X$}
         **for** each x **do**
            {update weight vector $\omega_k$ according to Equation (1)}
            Note: Including winner and neighborhood function
         **end for**
         Note: Not update $\alpha$ and $\sigma$ because are function of the epoch
      **end for**
      **return** $SOM^i = \omega_k^i \in X, k = 1...m * n$
   **end for**
   $S := \cup_i SOM^i$
   Apply Algorithm 1 1-layer SOM
   **return** $SOM = \omega_k \in X, k = 1...m * n$

---

## 3. The Computational Experiment

The SOM algorithm application has been extended to various fields: digital marketing [21], recommendation systems [22], visualization [23], and engineering [24]. The SOM allows a visualization of the interpretable sample, a flexible dimensional reduction that can give rise to good inferential models even in situations not previously evaluated.

In our experiment, we wanted to represent a limiting situation that sometimes occurs in some of these applications, a situation in which a small group of a population is nevertheless relevant and possibly narrowly bounded and deviates in some of its characteristics from the general context, being very similar in the rest, for example in the field of customer management of a bank, customer type profiles that vary from these general structures due to the effect of environmental components such as the place where they live. Another example is in the industrial field, in the management of raw materials, raw materials that occasionally undergo small changes in their basic characteristics due to low-impact weather variations.

We now describe the computational experiment, specifying:

1. The stochastic model used to generate the data in a higher (original) dimension (three in our case);
2. The stratified sampling procedure to generate the data;
3. The structure of the one- and two-layer SOMs used to summarize or cluster the structure in a reduced (2D) dimension map;
4. How the initialization of the map nodes was performed;
5. The measurement of the conservation of topology;
6. Results in the form of visualization.

### 3.1. The Stochastic Gaussian Mixture Model for the Simulations

The stochastic model used for the simulations was a mixture of 3D normal populations, defined by a covariance matrix with low values and a vector of means, selected to create a mesh in the 3D space.

$$x - \mu_k = \begin{bmatrix} x_1 - \mu_{1k} & x_2 - \mu_{2k} & x_3 - \mu_{3k} \end{bmatrix} \tag{5}$$

$$f(x) = \Sigma_{k=1}^{K} \frac{\psi_k}{(2\pi)^{3/2}\sqrt{M_k}} e^{-\frac{1}{2}(x-\mu)M_k^{-1}(x-\mu)^t}, \tag{6}$$

where $f$ is the density function of a multivariate Gaussian mixture, $\mu = \begin{bmatrix} \mu_{1k} & \mu_{2k} & \mu_{3k} \end{bmatrix}$ are the component mean vectors, $\psi_k$ are the component weights, and $M_k$ the component variance–covariance matrices. A six-component mixture was used with the mean and weights shown in Table 1, all lying on the hyperplane $x = z$. Figures 1 and 2 illustrate this.

Using the z-coordinate, we defined our outlier component and designed the simulation experiment with the following main features:

The outlier component was a low-weight clone of one of the other six components, but located in:

$$z = x - 0.1, \tag{7}$$

i.e., outside the hyperplane where the centroids of the remaining components lie, but very close.

**Table 1.** Gaussian mixture mean parameter definition.

| Component | X | Y | Z | $\psi$ |
|:---:|:---:|:---:|:---:|:---:|
| 1 | 0.88 | 0.71 | 0.88 | 3000 |
| 2 | 0.82 | 0.11 | 0.82 | 3000 |
| 3 | 0.46 | 0.21 | 0.46 | 3000 |
| 4 | 0.12 | 0.11 | 0.12 | 3000 |
| 5 | 0.19 | 0.81 | 0.19 | 3000 |
| 6 | 0.49 | 0.81 | 0.49 | 3000 |
| Outlier | 0.49 | 0.81 | 0.39 | 90 |

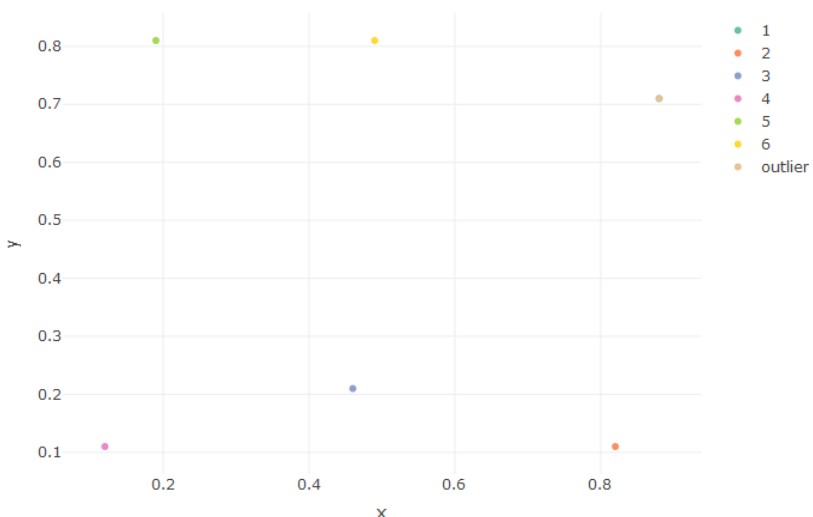

**Figure 1.** Mean vector of each component, in x, y coordinates.

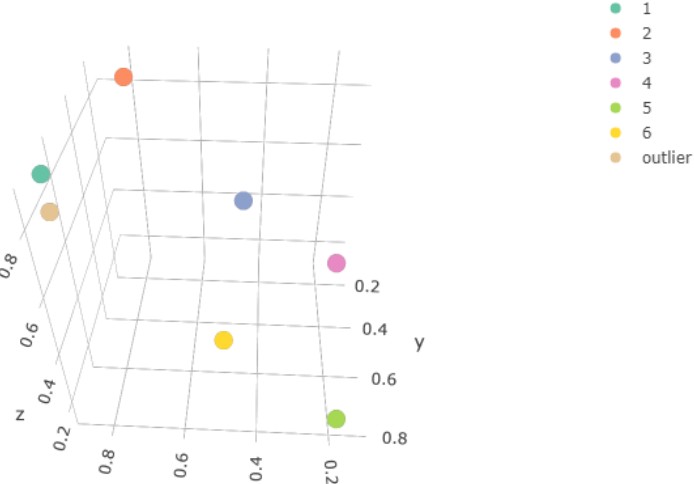

**Figure 2.** Mean vector of each component, in x, y, z coordinates.

As mentioned above, the variance–covariance matrices have the same low values for the six mixture components. Since z is equal to x (except for the outlier), Equation (8) represents the covariance matrix used by all of them for the x and y coordinates.

$$\sigma = \begin{pmatrix} 0.008 & 0.0064 \\ 0.0064 & 0.008 \end{pmatrix} \tag{8}$$

Figure 3, above, shows the distribution of the Gaussian components in xy-coordinates. The same covariance matrix generates similar densities, lying in different regions. We just found similarity between Component 1 and the outlier one, the only component with a lower frequency than the others. Figure 4, above, shows better how the outlier component lies on a parallel hyperplane, following Equation (7).

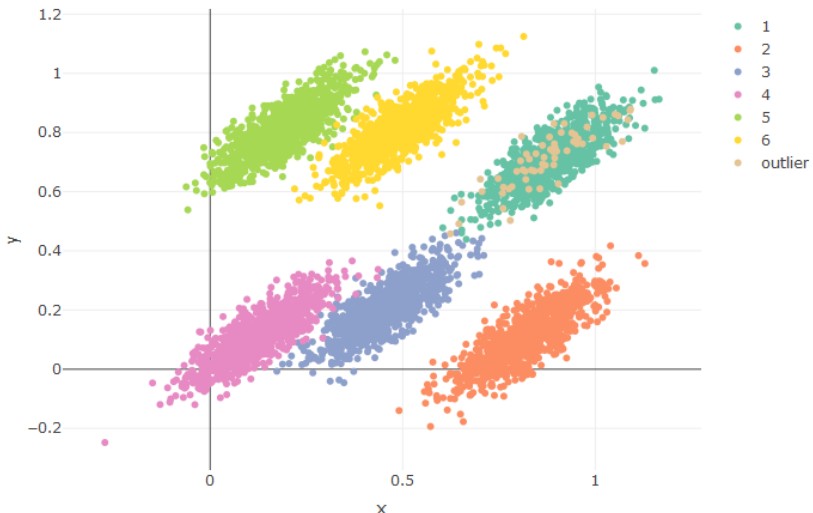

**Figure 3.** Generated components.

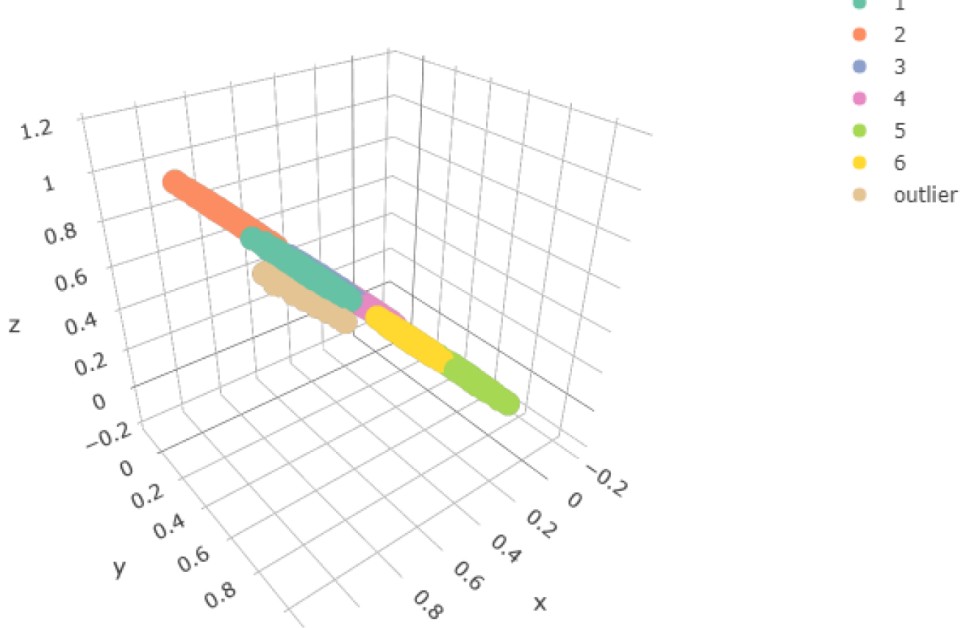

**Figure 4.** Population used in the experiment, x, y, z coordinates

### 3.2. The Sampling Procedure and Strata Configuration

Once we fixed the model parameters, we sampled from the data-generating process in accordance with the component weights. This sample would be the input data to the one-layer and two-layer SOMs, i.e., the same sample was be used for both cases. The size of the sample of each component was 3000 except for the outlier, for which it was 90; see Figure 4.

We defined different strata that simulated situations in which prior knowledge allowed us to segment into blocks with less unbalanced localized frequencies and fewer alternative components. The allocation of mixture components to strata is given in Table 2. Note that the relative frequencies of the components in some of the strata were significantly different from their global frequency.

**Table 2.** Allocation of mixture components to strata.

| Component/Strata | First | Second | Third | Fourth |
|:---:|:---:|:---:|:---:|:---:|
| 1 | 3000 | 0 | 0 | 0 |
| 2 | 0 | 1200 | 1200 | 600 |
| 3 | 2400 | 0 | 0 | 600 |
| 4 | 0 | 1500 | 1500 | 0 |
| 5 | 0 | 0 | 3000 | 0 |
| 6 | 3000 | 0 | 0 | 0 |
| Outlier | 0 | 45 | 0 | 45 |

The stratification was our means to elicit prior information, which should have some relevant content, but not be too specific. We proposed a situation in which we had incomplete a priori information on an alternative grouping of individuals in which the measured characteristics (x, y, z coordinates) were not taken into account. If it were specific, we would be dishonest by putting too much information, which in real problems is not available in that detail. Usually, we do have some information, but not conclusive.

For the simulation, we considered a nontrivial number of strata (higher than two). In a real situation, the outlier can be distributed in many different ways along the strata. For illustration, we selected one of them, the one specified in Table 2, in which some strata

did not include any outlier individual, some included the outlier with some (not very similar) other components, and another one where the outlier could be confused with other components because it was quite close to them. We worked with the minimum number of strata necessary to represent all three possibilities.

This situation can occur in various examples:

Medicine: such as the measurement of various symptoms (x, y, z coordinates) in patients at different stages of evolution of a disease (each group or mixture component) measured in different hospital centers (stratum). In each hospital (stratum), we can find patients at different stages of evolution (group), and each stage of evolution includes patients from different hospitals.

The outlier group corresponds to a small proportion of patients that have an extremely advanced stage of evolution. A more frequent group of patients has some similarities (symptoms x, y, x) with the outlying ones.

Marketing: clients on whom one measures different behavioral indicators (x, y, z) with a different state of engagement (each of the components of the mixtures or groups) extracted from samples stratified by premises with different locations (strata).

### 3.3. Structure and Parameters of One- and Two-Layer SOMs

The one-layer SOM was a $10 \times 10$ map. For the two-layer SOM, four $10 \times 10$ maps were built in the first layer, one for each stratum. Then, the 400 nodes of these maps were used as the sample for the second (final layer) SOM, which produced a $10 \times 10$ map as well, in such way that the comparison of the one-layer SOM was performed on equal footing. The topology in both cases was hexagonal.

Euclidean distances were used to select winning nodes. The values of the parameters were used for weight updating, in accordance with $\alpha_1 = 0.05$ and $\sigma_1 = 0.5$

### 3.4. SOM Node Initialization

Node initialization was performed in two ways:

1. Each of the four maps was initialized from the data in one of the four strata, i.e., one map per stratum; this would give equal weight in the initialization to all strata, regardless of their relative weights in the stochastic model;
2. Initialization taking into account the weights of the strata assigning numbers of nodes proportional to the sizes of the strata, i.e., strata with double the number of points assigned would have their data present in the initialization of twice the number of nodes.

## 4. Conservation of Topology

The SOM is intended to cluster the data and at the same time project them onto a lower dimensional space while preserving topology, i.e., in such way that points that are close in the original space belong to the same or close nodes (clusters) in the map and likewise (ideally) for those that are far apart or in-between.

The SOM has thus to pursue optimality in two simultaneous directions: cluster homogeneity and conservation of topology. It should be highlighted that a bicriterion optimization would result in being less optimal for the individual criteria than that which would be attained if only one criterion were pursued. As for cluster homogeneity, optimal means that clusters are as homogeneous internally as possible and, consequently, heterogeneous from one to the other. Thus, for the given total sample variability, within-cluster variability should be minimized and, consequently, between-cluster variability maximized. For instance, it was expected that k-means would be more successful in cluster homogeneity, given that it can focus on it without "having to worry" about the conservation of topology.

Second, in the preservation of topology, there are different perspectives: to preserve the topology and how to achieve this, although collaborative learning with the neighborhood structure always ensures a certain representativeness in this sense.

In our case, we chose the option of preserving the metric space, that is seeking to maintain the distances between the elements of the original space from their map centroids (weights). It is important to realize that dimension reduction makes a fully accurate representation impossible.

Given a sample of size n, there is an $n \times n$ symmetric distance matrix in the original space and a corresponding SOM distance matrix of equal size. Optimal would then mean minimizing the distance between these two distance matrices.

The distance matrix in the original space is rather straightforward: euclidean distances could be a reasonable solution, and they were, after all, the measure adopted for the SOM competition stage.

The SOM distances are more complex. To start with, if only distances between the node integer pairs in a 2D map were taken into account, then the node centroids would not be considered, and relevant information would be neglected. A sound criterion should thus take both distances into account, i.e., the 2D pairs of integers and the between-centroid (original space) ones. Introducing the structure of the map in the distance measurement is a challenge, since starting with the two distances would be on different scales, it also adds one more factor to take into account when combining both. Here, we used a distance defined in terms of the two (distances) and considered that not only the winning nodes of the points have to be taken into account, but also their neighborhoods. In a nutshell, we measured the distance between the neighborhoods of the corresponding winning nodes.

In order to take into account the topological structure of the map when measuring the validity of the SOM, we worked with two distances that took into account the weights and topological structure: image-based and graph-based.

*4.1. Image-Based Distance*

This is based on creating an image map for each of the two points $x_i, x_j$ involved; the dimensions of each image are the same as those of the SOM under study. The distance is a weighted average of the distances between the centroids of all possible pairs of nodes from the two images, the first element in the pair stemming from the $x_i$ image map and the second from the $x_j$ map. The weights of the average depend on the distance between the node in question and the winning node. Two possible weights were contemplated:

(a) The first would be the product of two factors, a monotonic function of the distance from the winning node to the node in question in the first ($x_i$) map, and the second likewise for the second ($x_j$) one;

(b) The second would be interpreted as the joint probability of the two nodes, which would depend on how the nodes are seen for each image map. This would involve the product of four factors, the first two for the weights of the first ($x_i$) node from the point of view of both images and the second likewise for the ($x_j$) node.

Here, we applied option (a).

The image-based distance, which came from [13], was thus formally defined as follows:

1. Let $x_i, x_j \in \mathbb{R}^n$ be the sample individuals:

   (a) $n_{pq} \in som_{map}$ the $pq$-th node in the map:

   (b) We call $\varpi_{pq} \in W \subset \mathbb{R}^n$ the centroids (projection-representative in the original space);

   (c) We call the neighborhood of $N_{pq}$ as $V(N_{pq}) = \{N_{rs}/d_2((p,q),(r,s)) <= \alpha\}$ with $p, q, r, s \in \mathbb{N}$;

   (d) We denote by $som_{map} = (NxN, W)$, where $N \subset \mathbb{N}, W \subset \mathbb{R}^n$ and $som_{map}(x_i) = (N_{V(N_i)}xN_{V(N_i)}, W_{V(N_i)})$, the projection of the original map onto the neighborhood space for the node assigned to point $x_i$;

   (e) A map may be considered an image, a pixel matrix where each pixel has been assigned the weight vector of the centroid of each node. We shall now lay out for our problem the modifications on distances used in such cases;

   (f) Let us define the global distance, which combines the spatial distance with the pixel intensity:

i. $GD(A, B) = \frac{2}{N^2(N^2-1)} \sum_{A_{ij}} \sum_{B_{lm}} \delta(A_{ij}, B_{lm})$;

ii. $d^D((i, j), (l, m)) = \frac{|i-l|+|j-m|}{N}$;

iii. $\delta(A_{ij}, B_{lm}) = \alpha d^D((i, j), (l, m)) + \beta GR(a_{ij}, b_{lm})$ with $0 \leq \alpha, \beta \leq 1$, $\alpha + \beta = 1$;

iv. $GR(a_{ij}, b_{lm}) = |a_{ij} - b_{lm}|/max(a_{ij}, b_{lm})$. This is the definition for black and white images. In our case, we worked in $W \subset \mathbb{R}^n$, so we replaced it with the normalized euclidean distance;

(g) Once a framework has been established, we specified image distances within the framework. We now considered the "images", the submaps of the neighborhood assigned to each point, and we assigned a variable to each image, an *I* intensity matrix that contains the values determined by the neighborhood:

i. $I / I_{ij}(x_i) = f(N_{ij}, N_{nl}) \in (0, 1)$, where $N_{n,l} = N(x_i)$;

ii. *f* may be defined in several ways, but here, we chose: Equation (9)

$$f(N_{mk}, N_{nl}) = \begin{cases} 1 \ winner \\ 0.75 \ first - level \ neighborhood \\ 0.5 \ second - level \\ 0.25 \ other \ cases \end{cases} \tag{9}$$

iii. We used the distance above to compare the two images, by means of the following transformation:

iv. We define $dNodes(N_{nl}, N_{mk}) = \lambda_{ij}(N_{nl}, N_{mk})\delta(N_{nl}, N_{mk})$, and then, $GD(SOM(x_i), SOM(x_j)) = \frac{2}{N^2(N^2-1)} \sum_{SOM_i} \sum_{SOM_j} dNodes_{ij}(A_{nk}, B_{lm})$;

v. $\lambda_{ij}(N_{mk}, N_{nl}) = I_{mk}(x_i)I_{nl}(x_j)$;

vi. A hierarchy within the distances is thus defined:

$$\lambda_{ij}(N_{mk}, N_{nl}) = \begin{cases} 1 \ both \ winners \\ 0.75 \ winner \ and \ 1 - level \\ 0.565 \ both \ 1 - level \ neighborhood \\ 0.5 \ winner \ and \ 2 - level \ neighborhood \\ 0.375 \ 1 - level \ and \ 2 - level \\ 0.2 \ both \ 2 - level \end{cases} \tag{10}$$

### 4.2. Graph-Based Distance

We took all paths that went from the $(x_i)$ winning node to the $(x_j)$ winning node and obtained the shortest one by application of the Kruskal algorithm [25]. For this path, we added up all the distances between the centroids of the consecutive nodes. Neighborhoods were needed for $(x_i)$ and $(x_j)$, since whether two adjacent nodes will be considered consecutive in a path depends on howthe neighborhood is defined (diagonal nodes could be considered nonconsecutive). Figure 5 illustrates this.

The graph-based distance is thus formally defined as follows:

Let $x_i, x_j \in \mathbb{R}^n$ be the individuals in the sample:

1. $n_{pq} \in som_{map}$ the *pq*-node in the map:

   (a) Let us call $\omega_{pq} \in \mathbb{R}^n$ the projection-representative in the original space; generally, it is approximated as: $\alpha < 1$,
   $\omega_{pq} = \sum_{x_i/N(x_i)=N_{pq}} x_i + \sum_{x_j/N(x_j)\in V(x_i)} \alpha x_j / card(V(N_{pq}))$;

   (b) The neighborhood of $N_{pq}$ is defined as follows $\alpha < 1$ :
   $\{N_{rs}/d_2((p, q), (r, s)) <= c\} p, q, r, s \in \mathbb{N}$;

2. Let $N(x_i) = \arg\min_{N_{pq}\in som_{map}}(d(x_i, w_{pq}))$;

3. Let $G = (V, E)$ be a nondirected graph, where $V = \{N \in som_{map}\}$ and $E = \{(N, M), N, M \in som_{map}/N \in V(M), M \in V(N)\}$;
4. We define for each side its weight $p(e_{(N,M)}) = d(N, M) = d(w_N, w_M)$;
5. Let us define a path from N to M as a sequence $(e_1, e_2, e_3, ...e_m)$ of sides such that $e_i$ ends where $e_{i+1}$ begins and $e_1, e_m$ begins or ends in N, M, respectively;
6. $\mathcal{C}_{\mathcal{NM}}$ the set of all paths beginning in N and ending in M;
7. By application of Kruskal's algorithm, one can find the minimal path from any node N to any other one, defined as $C_{NM} = argmin_{c \in \mathcal{C}}\{\sum_{e_i \in C} p(ei)\}$;
8. The distance between the points in the original space is thus: $d(x_i, x_j) = length(C_{N(x_i),N(x_J)})$.

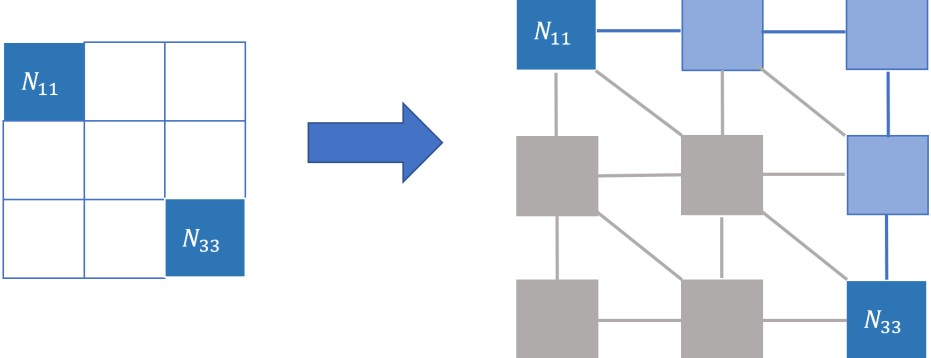

**Figure 5.** Graph distance. In this case, the distance comes from considering our map as a graph where the nodes are the nodes of the map and the edges represent the chosen neighborhood. The distance between winning nodes, representing the points in the new space, will be the minimum path in that graph, in this case, represented with the nodes and edges in blue.

### 4.3. Toy Example

Let us consider the following map where the 3D vectors in brackets are the weights (centroids) of the nodes.

$$\begin{bmatrix} (9,8,9) & (8,8,8) & (8,7,7) \\ (8,9,8) & (8,7,8) & (7,8,6) \\ (7,7,7) & (7,6,5) & (5,5,5) \end{bmatrix} \tag{11}$$

Let us compute the distances between two points $x_i$ and $x_j$ such that their winning nodes are (1,1) (upper left) and (3,3) (lower right), respectively.

### 4.3.1. Image Distance

For node (1,1), the weight matrix is:

$$\begin{bmatrix} 1 & 0.75 & 0.5 \\ 0.75 & 0.5 & 0.25 \\ 0.5 & 0.25 & 0.25 \end{bmatrix}$$

and likewise for (3,3):

$$\begin{bmatrix} 0.25 & 0.25 & 0.5 \\ 0.25 & 0.5 & 0.75 \\ 0.5 & 0.75 & 1 \end{bmatrix}$$

The $\delta$ matrix for all nine combinations of nodes in both maps is:

Translating the point distance to the map space as follows:

$$d^D(map, map) = \tfrac{1}{N}\left[d^D((i,j),(l,m))\right]_{(i,j),(l,m)}$$

$$d^D(map, map) = \frac{1}{9}$$

|       | (1,1) | (1,2) | (1,3) | (2,1) | (2,2) | (2,3) | (3,1) | (3,2) | (3,3) |
|-------|-------|-------|-------|-------|-------|-------|-------|-------|-------|
| (1,1) | 0 | 1 | 2 | 1 | 2 | 3 | 2 | 3 | 4 |
| (1,2) |   | 0 | 1 | 2 | 1 | 2 | 3 | 2 | 3 |
| (1,3) |   |   | 0 | 1 | 2 | 1 | 2 | 3 | 2 |
| (2,1) |   |   |   | 0 | 1 | 2 | 1 | 2 | 3 |
| (2,2) |   |   |   |   | 0 | 1 | 2 | 1 | 2 |
| (2,3) |   |   |   |   |   | 0 | 1 | 2 | 1 |
| (3,1) |   |   |   |   |   |   | 0 | 1 | 2 |
| (3,2) |   |   |   |   |   |   |   | 0 | 1 |
| (3,3) |   |   |   |   |   |   |   |   | 0 |

$$d(map, map) =$$

|       | (1,1) | (1,2) | (1,3) | (2,1) | (2,2) | (2,3) | (3,1) | (3,2) | (3,3) |
|-------|-------|-------|-------|-------|-------|-------|-------|-------|-------|
| (1,1) | 0 | 1.41 | 2.44 | 1.73 | 1.73 | 3.6 | 3 | 4.89 | 6.4 |
| (1,2) |   | 0 | 1.41 | 1 | 1 | 2.23 | 1.73 | 3.74 | 5.19 |
| (1,3) |   |   | 0 | 2.23 | 1 | 1.73 | 1 | 2.45 | 4.13 |
| (2,1) |   |   |   | 0 | 2 | 2 | 2.45 | 4.35 | 5.83 |
| (2,2) |   |   |   |   | 0 | 2.45 | 1.41 | 3.31 | 4.69 |
| (2,3) |   |   |   |   |   | 0 | 1.41 | 2.23 | 3.74 |
| (3,1) |   |   |   |   |   |   | 0 | 2.23 | 3.46 |
| (3,2) |   |   |   |   |   |   |   | 0 | 2.23 |
| (3,3) |   |   |   |   |   |   |   |   | 0 |

The expression for the distance is:

$GD(som_{11}, som_{33}) = \lambda_{11}(N_{11}, N_{12}) * \lambda_{33}(N_{11}, N_{12}) * \delta(N_{11}, N_{12}) +$
$+ \lambda_{11}(N_{11}, N_{13}) * \lambda_{33}(N_{11}, N_{13}) * \delta(N_{11}, N_{13}) + \ldots$
$+ \lambda_{11}(N_{11}, N_{33}) * \lambda_{33}(N_{11}, N_{33}) * \delta(N_{11}, N_{33}) + \ldots$
$+ \lambda_{11}(N_{33}, N_{33}) * \lambda_{33}(N_{33}, N_{33}) * \delta(N_{33}, N_{33})$
$= 1 * 0.25 * (\alpha 0 + \beta 0) + 0.75 * 0.25 * (\alpha 2 + \beta 2.44) + \ldots$
$\cdots + 1 * 1 * (\alpha 4 + \beta 6.4) + \cdots + 1 * 0.25 * (\alpha 0 + \beta 0)$

Figure 6 schematizes the example described.

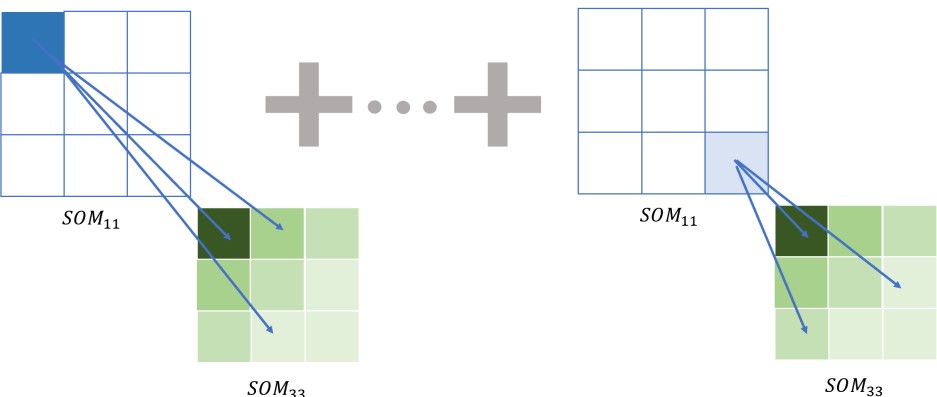

**Figure 6.** Image distance. In this graph, we summarize the idea of the metric. We measure the distance between the maps associated with each point. It will generally be the same map except for the winning node and its neighborhood. Here, we can observe that when comparing the winning node N11 of the SOM11 map (node in intense blue) with the nodes of the SOM33 map whose winning node is 33, we have to take this spatial relationship into account. The gradient of the colors represents the reduction of the effect of the distance between these nodes in the global distance, which is calculated with the lambda function.

### 4.3.2. Graph Distance

We obtained, for illustration, the distance for one of all possible paths between the two points. The Kruskal algorithm selects, within all possible paths, the one that gives the smallest distance and that will be taken as the final graph distance between the two points.

The neighborhood structure determines if, for instance, the path (1,1), (2,2), (3,3) is valid or if, on the contrary, going from (1,1) to (2,2), one should first pass through (1,2) or (2,1).

Let us choose the following path:

$$P = \{(1,1),\ (1,2),\ (1,2),(1,3),(2,3),(3,3)\}.$$
$$d_{graph}(x_i, x_j) = d_{graph}(map(1,1), map(3,3)) = \sum_{v_i \in P} d(v_i, v_{i+1})$$
$$d_{graph}((9,8,9),(5,5,5)) = d((9,8,9),(8,8,8)) + d((8,8,8),(8,7,7)) + \ldots$$
$$+ d((8,7,7),(7,8,6)) + d((7,8,6),(5,5,5)) =$$
$$= 1.4142 + 1.4142 + 1.73 + 3.74 = 8.2984$$

## 5. Results

The results of the full simulation exercise are given separately for the one-layer and two-layer maps and then compared.

The component centroids are plotted in Section 3: The Computational Experiment, and it can be observed that the first six lied in a plane, while the outlier center was outside it.

The sample points are represented in Figure 4, where, since the variances of the three variables in the components were small, the clouds of the different components are relatively tightly displayed along the plane.

### 5.1. One-Layer SOM

As mentioned above, the number of iterations was 2000, i.e., we ran the algorithm over the full dataset 2000 times.

Through the following figures, we can understand its application and draw different conclusions.

In Figure 7, we represent the values of each of the coordinates of the centroids of the nodes of the resulting SOM using a color scale. Notice that the maps referring to the x (first) and z (third) coordinates match exactly.

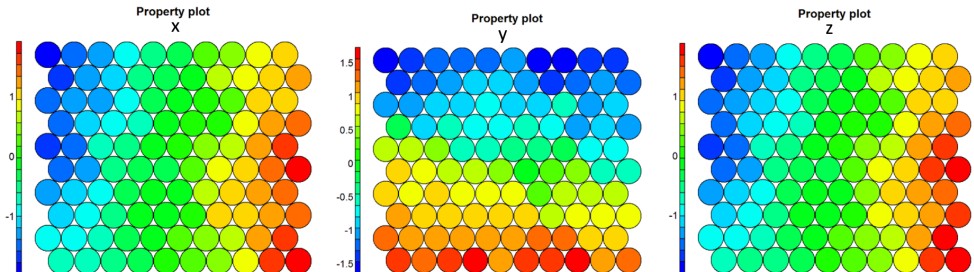

**Figure 7.** 1-layer Final SOMs for coordinates x, y, z, respectively.

Figure 8 represents in the three-dimensional space x, y, z (original space) the mean value of each of the mixtures together with the weights of the different nodes of the one-layer map. Notice how the only mean that is not overlapped by the centroids is that of the outlier component. It is the only component not correctly represented.

Figure 9 shows all the points belonging to the outlier component and the centroids of the nodes of the one-layer map. Again, we observe how there is no type of overlap.

The components included in the hyperplane $z = x$ were well represented in the SOM, and there were many nodes whose weights were located in the original space close to the component means.

However, our outlier component, due to its proximity to one of the components included in the hyperplane $z = x$, was not represented by any of the nodes Figure 8. This was also observed when comparing the original $x, y, z$ spaces' SOM property plots, which represent each of the nodes in the map, as shown in Figure 7: The outlier values were not represented in the weights of the map nodes, as expected. This behavior is particularly striking in the z-property plot.

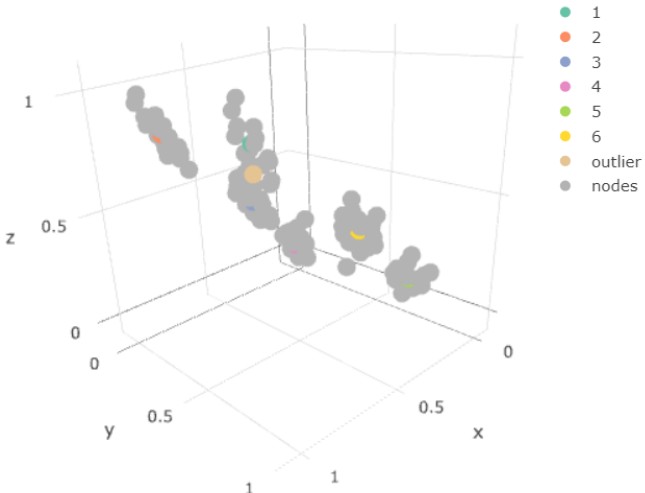

**Figure 8.** 1-layer Final SOMs for the original spaces.

In Figure 8, the node weights of the one-layer SOM were added to the component mean plot, confirming the above-mentioned situation. We observe in Figure 9 that the sample of points of the outlier component did not coincide with any of the weights obtained.

In Figure 10, we can see in three dimensions the points of the sample together with the weights of the nodes obtained (node tag), eliminating the component closest to the outliers. We validated the statement again by checking that the weights were located in the area of the component, the closest to the outlier component, but no weights corresponded to the outlier itself.

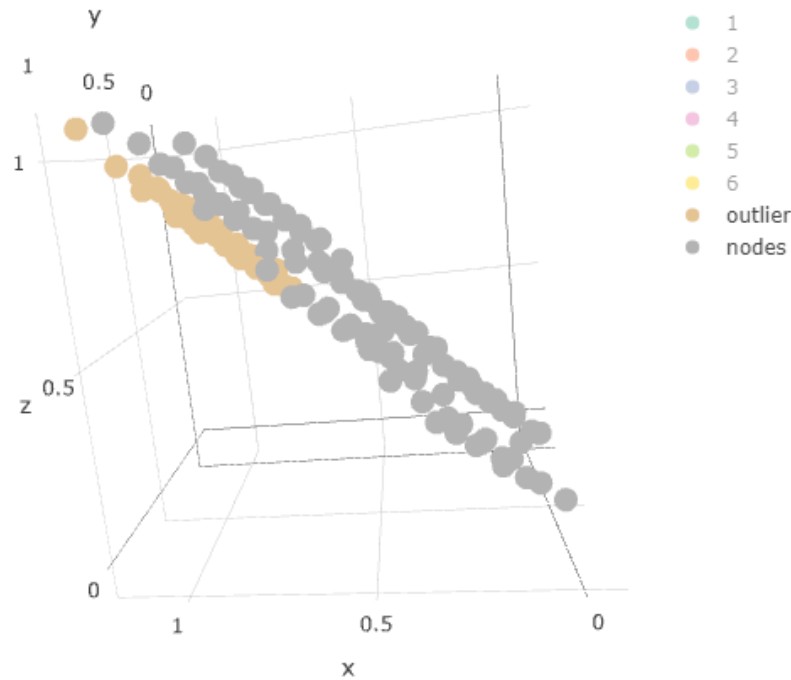

**Figure 9.** Weights over outlier samples.

In Figure 11, we represent the variable the x and y coordinates of the population together with the x and y coordinates of the nodes obtained. We show the projection of Figure 8 onto the xy plane. It was observed that the outlier points were distributed more or less evenly along the remaining components' points projections, which obviously did not include the z coordinate.

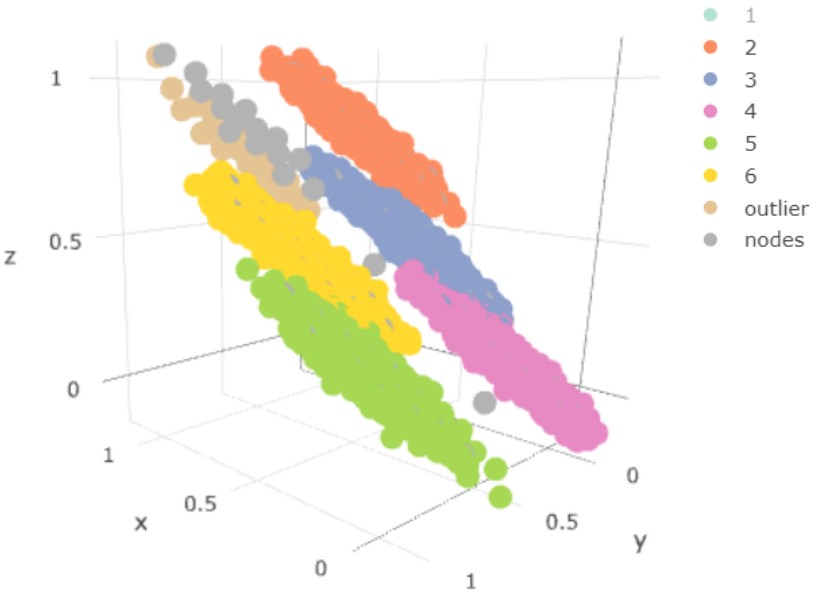

**Figure 10.** Three-dimensional representation of the data obtained and the weights, without the sample of the component.

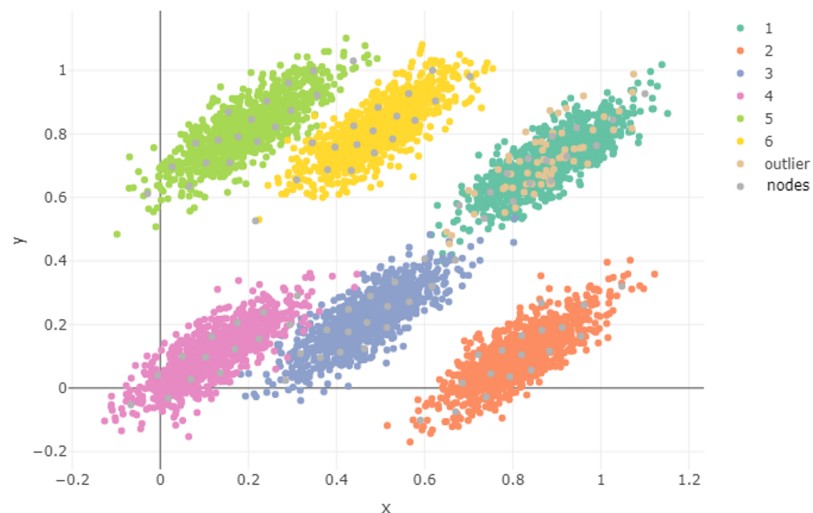

**Figure 11.** Projection.

### 5.2. Two-Layer SOM

We defined in Section 2 the same distribution of components for all strata. Each stratum included values from three components, and each component was included in more than one stratum. The frequency of appearance of each of the components in each stratum was similar except in the second one, where the outlier component maintained the overall frequency rate.

#### 5.2.1. First (Intermediate) Layer Results

Each map was generated using one stratum as the input. In each case, all components were well represented and kept the topology, because intermediate centroids appeared between far-apart components.

If we focused on the two intermediate maps where the outlier component was included, we may observe that if the relative frequencies of the outlier and the remaining components were balanced, there was a larger number of nodes in the map with nearby

centroids in the original space. This increased the probability that they appeared in the last layer.

To illustrate this, we show results for some of the strata.

In the case of Figure 12, the first stratum did not contain any sample point of the outlier component. However, it is interesting to view how it generated a smoother transition. The points in the "circle" in blue in Figure 13 are the map nodes that were generated to maintain the topology of the system, establishing, as pointed out by [17,18], the distance.

In Figure 14, we show the node weights for Stratum 4, which contained data from the 2, 3 components and the outlier component. Note that the outlier points now appeared, as opposed to the situation in the single-layer map, but also that intermediate points were generated to make the transition smoother. This gave us a clue about what would happen to an even more separate outlier.

In Figure 15, we show the 2D projection of the first-layer SOM for Stratum 4; the points are scattered around two straight lines because the outlier points are aligned with those of Component 3; the orange one (outlier centroid) is in (0.46, 0.21); it is more difficult for it to move "continuously" from green to orange or green to blue than from blue to orange where there is continuity.

In Figure 16, we show the property plots for the first-layer SOM of Stratum 4; note the z plot where the outlier nodes are on the lower right corner. We obtained what we expected: by reducing the sample size and keeping only some of the less neighboring components, it became easier to identify the elements of the low-frequency component. Let us look at how this extended to the second layer.

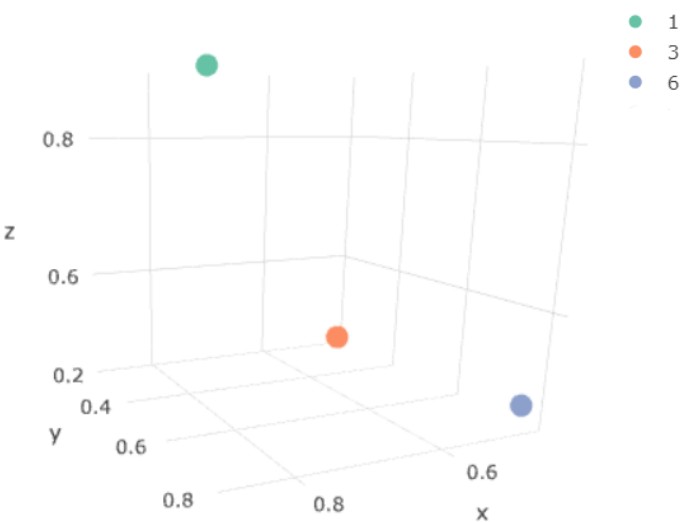

**Figure 12.** The centroids of the first-layer SOM for the first stratum.

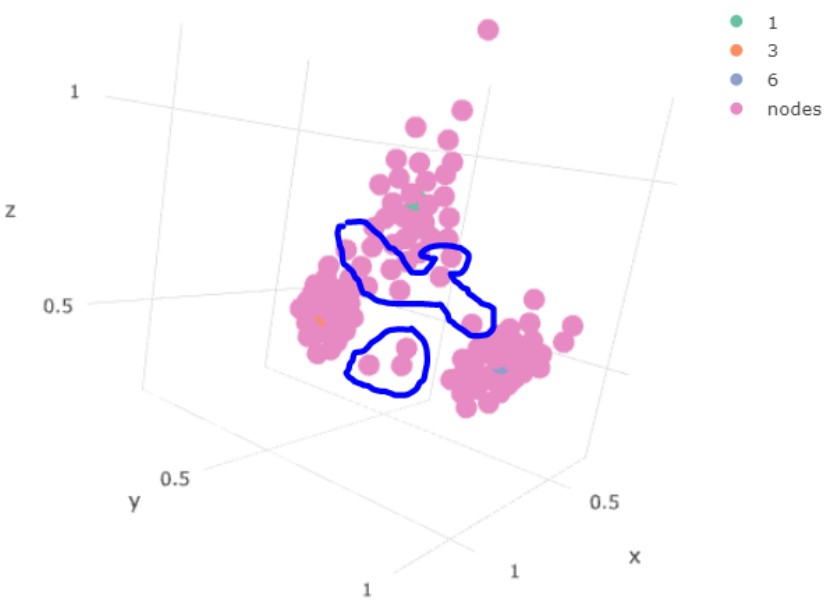

**Figure 13.** Status of the weights associated with the map generated for the first stratum.

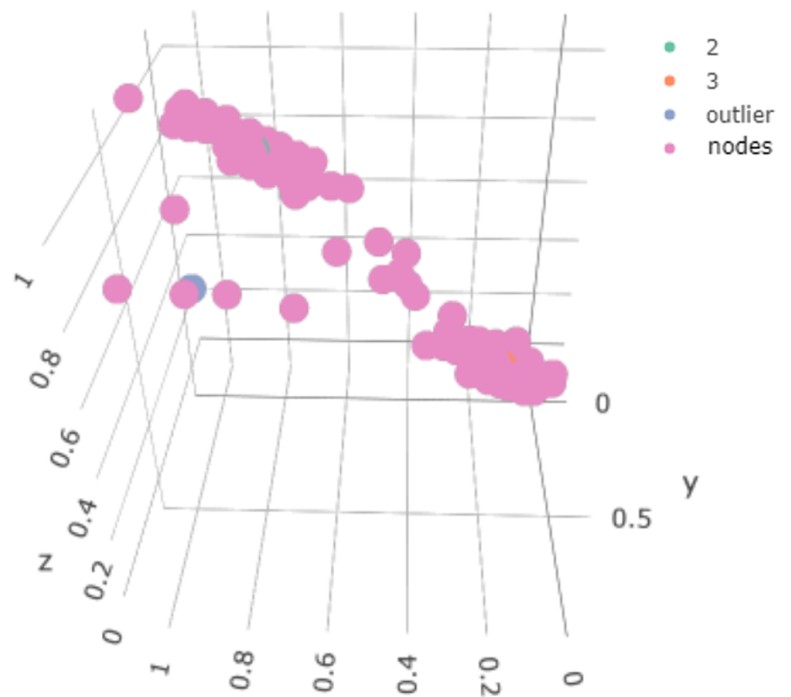

**Figure 14.** Status of the weights associated with the map generated for the second stratum.

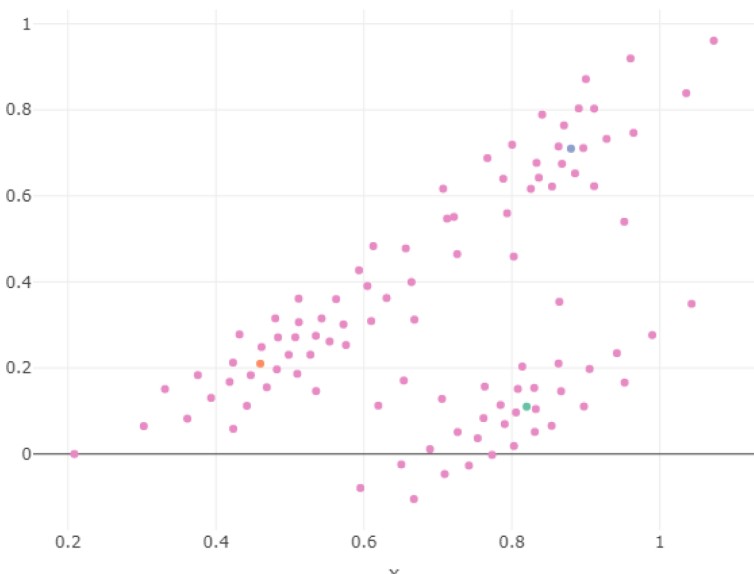

**Figure 15.** Two-dimensional plot of the weights in the second stratum SOM.

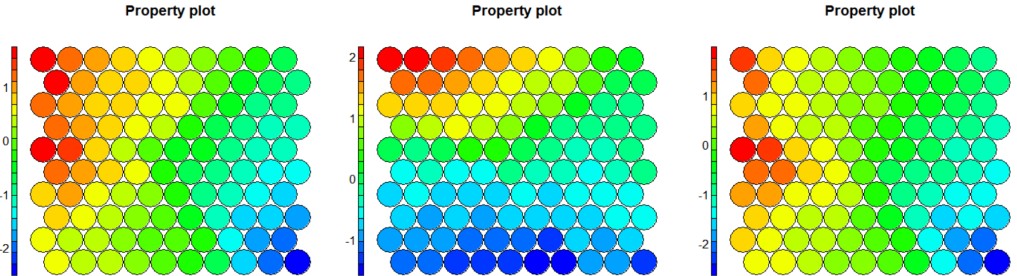

**Figure 16.** Property plots of Stratum 4.

### 5.2.2. Second-Layer Results

The results of this last layer substantially improved the single-layer alternative. We verified that the outlier component was well represented; several centroids of the map nodes were located in the original space in the density zone of the outlier component.

We show the property graphs for the final SOM layer in Figure 17: if we focus on the z graph, we can observe the presence of the outlier nodes, as well as the relatively smoother transition than in the one-layer case of the atypical nodes to those of the remaining components. In Figure 18, we verify that, now, some of the map weights were close to the centroid of the outlier component. This made this model more representative of the situation in the original space, and this can also be observed in the results from the SOM metrics in Section 5.3.

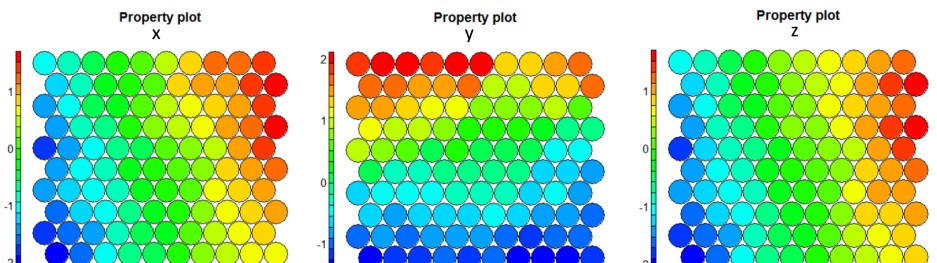

**Figure 17.** Two-layer SOM, last layer, final maps for coordinates x, y, z, respectively.

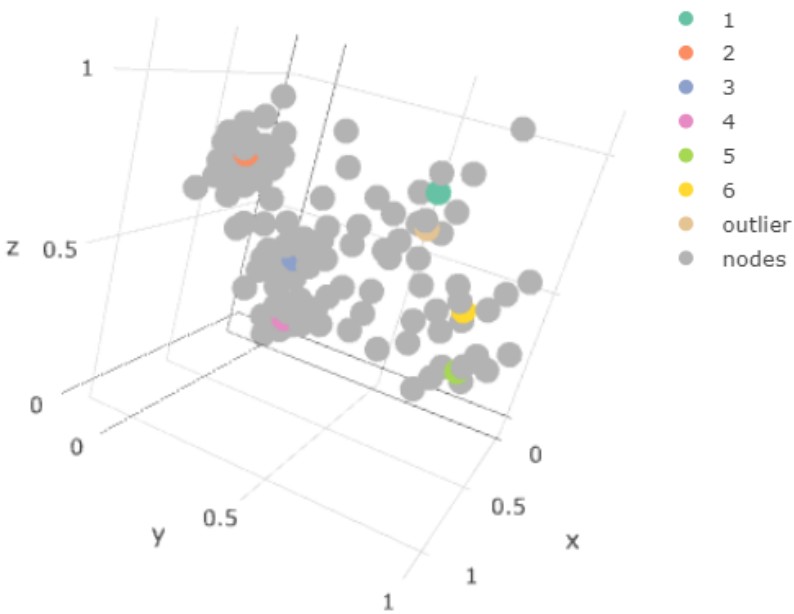

**Figure 18.** Final 3D result.

*5.3. Conservation of Topology*

In Table 3, we show the image and graph distances for the one- and two-layer SOM in the two examples.

We sought to validate the preservation of the topological space by checking that the metric space was maintained. We checked that the distances between the points in the original space were preserved when projecting them on the map. We show the distance between the matrices of the distances between the points of the original space and the matrix of the distances between their representatives on the SOM in the original space. We used the euclidean distance and, to measure the distance between the weights (representatives), the graph-based metrics (where we considered the nodes linked to their neighborhood as representative) or on images (where we considered the map with its topological structure as representative).

This is one of the greatest properties of the SOM compared to other clustering algorithms such as K-means, DBSCAN, or dimensional reduction such as T-SNE or UMAP [26]. The main difference is to introduce "fictitious" areas on the map, to which there are no assigned points in the original space, to preserve topology. In this way, it is more robust when making inferences based on topology even on points in space that, however, have not yet appeared in the sample.

**Table 3.** Table of the results of the different distance metrics of the algorithms used.

| Model | Example | Image Distance | Graph Distance |
|---|---|---|---|
| 1-layer SOM | 1 | 32,456 | 5032 |
| 2-layer SOM | 1 | 22,456 | 4625 |
| 1-layer SOM | 2 | 22,456 | 4048 |
| 2-layer SOM | 2 | 17,456 | 3740 |

## 6. Interpretation of the Results and Concluding Discussion

The classic single-layer model obviates low-frequency components located close to similar higher frequency ones.

That is why we developed the multilayer alternative with data segmentation in layers, the two-layer SOM. If it adequately represents the total set of points in the sample in addition to maintaining a better topological structure, it is a valid solution for its application

in various areas where we can find these disparate low-frequency components, even more so if it can only be fuzzily delimited.

We encountered several challenges. First was to define a stochastic model through which one can illustrate the contribution of our methodology, for example the low-frequency group of the population needed to be represented because, in the one-layer strategy, it was absorbed by higher frequency neighboring groups. Second was that an adequate stratified sampling procedure had to be developed. Some, but not excessive, prior information was required for adequate outlier group detection. Third, the computational complexity had to be controlled; parallelization was essential to this end. Fourth was the definition of one or more adequate distance measures to quantify conservation of topology: traditional distances measures were not satisfactory for our specific problem because some low-frequency regions of the population were not accounted for.

From the results obtained in this article, it can be concluded that:

1. The classic single-layer model ignored low-frequency components located between similar higher frequency ones. This was reflected in the distance matrix, as can be seen in the developed metrics. One was not adequately representing the original space or translating its topology. Not only was one ignoring one set of the sample, but this affected the representation of other points in the original space;

2. The stratification of the data (which may be given by prior knowledge) allowed, in two-layer SOMs, generating a map in which these components were also well represented;

3. For the two-layer map, stratification would result in the generation of intermediate-layer nodes that would represent points lying between components far apart, but included in the same stratum, and as inputs to the last layer, they would end up appearing to generate a more representative map, also in a topological sense.

Therefore, the methodology developed for the bilayer SOM was proposed to solve the problem of the representativeness of low-frequency elements that can be assigned to other components of the system, as well as a better strategy to maintain the topological structure understood as distances.

**Author Contributions:** Conceptualization, G.A.V.C., J.M.M.M., and B.G.-P.; methodology, G.A.V.C., J.M.M.M., and B.G.-P.; software, G.A.V.C.; validation, G.A.V.C.; formal analysis, G.A.V.C. and J.M.M.M.; investigation, G.A.V.C., J.M.M.M., and B.G.-P.; writing—original draft preparation, G.A.V.C., J.M.M.M., and B.G.-P.; writing—review and editing, G.A.V.C., J.M.M.M., and B.G.-P.; visualization, G.A.V.C.; supervision, G.A.V.C., J.M.M.M., and B.G.-P.; project administration, G.A.V.C. All authors read and agreed to the published version of the manuscript.

**Funding:** This research received no external funding

**Institutional Review Board Statement:** Not applicable.

**Informed Consent Statement:** Not applicable.

**Conflicts of Interest:** The authors declare no conflict of interest.

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
