# Peer review of "One-Layer vs. Two-Layer SOM in the Context of Outlier Identification: A Simulation Study"

_applsci, doi:10.3390/app11146241_

Round 1
Reviewer 1 Report
The authors aim to prove that SOM model is not biased by the way the data is distributed. The authors develop bilayer SOM model to improve outlier detection. The structure of the paper is reasonable, formulas and calculations are correct.
- The contribution of the paper is written in different sections. I suggest to add a sub-section from the introduction to clearly state the contributions of the paper and why it is important for SOM applications in multiple sectors.
- There is no mention to figure 7 in the text. It's better to make a brief explanation and analysis for every figure. Also, I noticed in figure 9, you mentioned that “We can see how any weights represents this points.” Put that inside text while explaining figure 9 not in the figure caption.
- The document contains a lot of formatting issues. For example the gap between figure 10 and figure 11. Please revise the formatting based on the journal requirements.
- Is there any recommendations or challenges faced the authors while conducting the experiments? Please mention them in the conclusion section.
- When explaining SOM, you keep repeating the same sentence from lines 106 to 110. Please revise. “SOMs show a self organizing behavior with the capacity of detecting hidden characteristics within non-labelled groups when enough map nodes (also called neurons) are used.With SOM(unsupervised), we are able to identify features and structures in high-dimensional data. SOMs display self-organizing behavior with the ability to detect hidden features within unlabeled groups when enough map nodes (also called neurons) are used”.

Author Response
"Please see the attachment."

Reviewer 2 Report
General considerations
This paper performs a study on the outlier detection capability of a SOM. They compare this capability with a classical SOM and a two-layer SOM that the authors propose. The practical application of this proposal is with pattern that have small group of population in which a feature deviates in some of its characteristics.
Although the extension proposed in this work allows the application of a variation of the SOM algorithm for the detection of outliers that meet certain conditions, I believe that the authors' proposal eliminates one of the great advantages of a SOM: its simplicity when training the network. The authors' proposal depends on several hyperparameters and distances (image distance and graph distance) that make it difficult to use and evaluate results in the application of this neural network to real problems.
However, I think that the proposal of this work can be considered an interesting contribution for the subset of problems in which there are outliers characterized by out-of-range values in one of its components.
Things that need to be clarified or corrected
Schemes 1 and 3 do not specify when the a(t) s(t) parameters are updated.
Equation (6) is not clear what it is used for. It indicates f(x1, x2), but then in the body of the equation x appears. What is the function f(x1, x2) used for? This should be corrected, and referenced in the text.
Lines 105 and 106 indicate that " A 6-component mixture was used with 196 mean and weights shown in Table 1" but this table only seems to show the average values.
The title of figure 2 should be “Mean vector of each component, in x, y, z coordinates”
It should be clarified whether equation (8) is the covariance matrix for x and y (since z=x).
I do not understand (I think it is not explained in the text) the information provided in table 2. Does it indicate that the set of patterns is divided into 4 parts, and then, for example, for group 1, 3000 patterns are taken from class 1, 2400 from class 3, 3000 from class 6 and no outliers? Why 4 groups? How have the number of patterns from each class been selected? Why only groups 2 and 4 have outliers? This should be well explained.
In 228 the initial values for a(t) and s(t) are given, but a0 and s0 do not appear in equations (2) and (4).
I do not understand point 2 in lines 234 to 236. What is the meaning of “strata with double weight”? This should be clarified.
The following text of lines 247 to 249 “Thus, for given total sample variability, within cluster variability should be maximized and consequently, between cluster variability maximized.” Doesn't this phrase contradict the homogeneity of the clusters previously mentioned?
Results
In lines 401 to 405 it is noted that the components of the outliers are not represented by any of the SOM nodes. How is this conclusion reached, because of the similarity between the x and z maps in Figure 7? Where on the map is this information?
Figure 8 and 9. What is the meaning of legend “salida”? How are these figures built?
The conclusion explained in lines 430 to 433 depends on the frequency of occurrence of the outliers being comparable to the frequency of occurrence of the rest of the features. Isn't this contradictory to the outlier search, a problem in which the frequency of out-of-range data is generally assumed to be much lower than that of the rest?
In the 2-layer SOM, if I understood the procedure correctly, using outliers with a higher frequency of occurrence than in the SOM-1 case results in nodes representing outliers. Does this mean that the ability of the proposed model to find outliers depends on how the strata has been obtained?
As before, the authors should explain how looking at Figure 17 leads to the conclusion that in the map for z the outliers are included.
Errata
71 – gthe - The
73 – each – Each
83 – Missing session number
The text on lines 109 and 110 is very similar to the text on lines 106 and 107, and they contribute nothing. Rather, it appears that the information is duplicated.
137 – a2 does not appear in the equation
138 – Separate equation 3
174 – Of – of
222 – “four 10x10maps” should be “four 10x10 maps”
334 – “swill” should be “will”
Caption of figure 5 need revision. Perhaps it needs a colon after “Image distance”.
405- Figure 8??
425 – “input,” should be “input.”
437 and 438 – “The points in "circle" in blue Figure 13 are map nodes that have been generated to maintain the topology of the system, establishing, as pointed out by [17] [18], distance.” Should be rewritten.
465 – “In the table below Table 3, …” could be changed by “In table 3, …”
Author Response
"Please see the attachment."

Round 2
Reviewer 2 Report
I would like to thank the authors for the clarifications and improvements they have made to this work based on my review.
After reviewing the modifications to the article, together with the authors' comments, I consider that all the doubts I raised in the review have been adequately clarified, and I consider that the work is now easier to understand. I consider, therefore, that all my considerations and modifications have been adequately addressed in this revision.